# Application and Selection of Remediation Technology for OCPs-Contaminated Sites by Decision-Making Methods

**DOI:** 10.3390/ijerph16111888

**Published:** 2019-05-28

**Authors:** Junping Tian, Zheng Huo, Fengjiao Ma, Xing Gao, Yanbin Wu

**Affiliations:** 1GIS Big Data Platform for Socio-Economy in Hebei, Shijiazhuang 050061, Hebei, China; tianping5566@126.com (J.T.); mafengjiao@heuet.edu.cn (F.M.); gaoxing@heuet.edu.cn (X.G.); 2School of Information Technology, Hebei University of Economics and Business, Shijiazhuang 050061, Hebei, China; huozheng@heuet.edu.cn; 3School of Management Science and Engineering, Hebei University of Economics and Business, Shijiazhuang 050061, Hebei, China; 4School of Public Administration, Hebei University of Economics and Business, Shijiazhuang 050061, Hebei, China

**Keywords:** organochlorine pesticides, contaminated site remediation, analytic hierarchy process, technique for order preference by similarity to ideal solution, persistent organic pollutants

## Abstract

The production and use of organochlorine pesticides (OCPs) for agricultural and industrial applications result in high levels of their residues, posing a significant risk to environmental and human health. At present, there are many techniques for OCP-contaminated soil remediation. However, the remediation of contaminated sites may suffer from a series of problems, such as a long recovery cycle, high costs, and secondary pollution, all of which could affect land redevelopment and reuse. Therefore, the selection of an appropriate technology is crucial for contaminated sites. In order to improve and support decision-making for the selection of remediation techniques, we provide a decision-making strategy for the screening of remediation techniques of OCP-contaminated sites. The screening procedure is proposed based on combining the analytic hierarchy process (AHP) and the technique for order preference by similarity to ideal solution (TOPSIS). The screening indexes include economic indicator, environmental indicator, and technical indicator. The assessment results show that co-processing in cement kiln obtained the highest overall score and was thus considered to be the most sustainable option. This suggested remediation technology was similar to the practical remediation project, indicating that the screening method could be applied for the selection of remediation technologies for sites contaminated with persistent organic pollutants.

## 1. Introduction

Organochlorine pesticides (OCPs) are a kind of persistent organic pollutant (POP) that are ubiquitous in the environment. Dichlorodiphenyltrichloroethane (DDT) and hexachlorocyclohexane (HCH) are the two major POPs that require priority control [1,2]. China has been a major producer and consumer of OCPs, and the total output of the two in question was 460,000 tons and 4.9 million tons, respectively, accounting for 20% and 33% of the world’s production [3,4,5,6,7]. Although this type of pesticide has been banned in China since the 1980s, there are still small amounts of DDT and HCH being produced as raw materials for chemicals or export demand [8,9,10,11]. The abuse of DDT and HCH in agricultural and industrial application promotes high levels of their residues, posing a major environmental and human health threat [12,13,14,15,16]. Although the area of OCP polluted industrial land is smaller than that of agricultural land, the impact of industrial-related OCP pollution is greater. Meanwhile, some contaminated sites with good geographical location where industrial enterprises have been shut down or relocated are in great demand for redevelopment due to the rapid urbanization and the implementation of ecological civilization construction in China [17,18]. Direct reuse of such contaminated sites without any remediation treatment could cause prominent environmental safety and human health risks [19,20,21,22,23,24]. Therefore, soil remediation techniques are urgently needed for further land re-utilization and redevelopment.

There are many kinds of remediation technologies for POPs-contaminated sites, including gas phase extraction [25,26,27,28,29], soil leaching [30,31,32,33], vitrification [34,35], chemical oxidation [36,37,38,39], incineration method [40,41,42,43], cement kiln co-disposal method [44,45,46,47], thermal desorption [48,49,50], and bioremediation [51,52,53]. The principle is either to increase the usability of contaminated soil through physical and chemical methods, or to degrade the contaminating chemicals through chemical or biological methods. Since remediation technologies may vary greatly in terms of processing efficiency, economic benefits, and environmental benefits, there may be difficulties in selecting the remediation technology that is best suited for each situation [54,55,56]. Therefore, the screening process plays an important role in selecting the most suitable remediation technology for contaminated sites. The screening of remediation techniques for OCP-contaminated sites has commonly been evaluated by experienced experts, although there may be different interests and perspective between technology developers and environmental authorities on the remedial solutions. Finally, different technologies are often selected because they are familiar but not because they are the most applicable or cost-effective for a given site. Additionally, contaminated sites may exhibit regional characteristics [57,58]. Thus, it is necessary to explore efficient decision-making method in order to select an appropriate remediation technology that can take account various factors affecting the decision-making process, such as economic factors, technical factors, social factors, etc.

Multiple criteria decision-making methods are commonly applied for the screening of remediation techniques, by which alternative remediation technologies could be evaluated comprehensively to obtain the optimal one [59,60,61,62]. There are several decision analysis methods including SAW (simple additive weighting) [63,64,65], OWA (ordered weighted average) [66,67], AHP (analytic hierarchy process) [68,69,70], PROMETHEE (preference ranking organization method for enrichment evaluation) [71,72,73], and TOPSIS (technique for order preference by similarity to ideal solution) [74,75]. SAW, OWA, and AHP use score-weighted summations to determine solutions for decision-making problems, although some subjective factors can impact the evaluation results. PROMETHEE utilizes the preference function, the criterion value, and the criterion weight to make decisions. Nevertheless, it can be restricted by the knowledge, experience, and preferences of the decision maker or expert. Since human judgments are often vague under many conditions in practice, the TOPSIS method cannot fully reflect the positional relationship of the scheme. There may be the problem that both the positive ideal solutions and the negative ideal solutions have similar relations [76,77].

In order to obtain scientific and reasonable decisions, it is usually necessary to establish and utilize a hybrid of several methods. Bai et al [78] developed an interactive method through combining AHP and TOPSIS for the screening of heavy metal-contaminated soil remediation techniques. Zhang et al [79] combined AHP and TOPSIS to screen the remediation techniques of organic contaminated soil in a coking plant located in an industrial zone in North China. The results are similar to those used in the practical remediation approach. These investigations provide a theoretical basis for technology selection by using an integrated multi-attribute decision-making methodology for OCP-contaminated site, although there are few works on it. Meanwhile, remediation technology with less secondary pollution would be preferred for decision makers due to the requirement for sustainable land use and development. Therefore, the environmental impact of remediation technology could be assessed typically. In this paper, we present a novel multi-criteria decision-making model based on the combination of AHP-TOPSIS for screening of the remediation techniques of OCP-contaminated sites. In order to obtain the appropriate remediation techniques, several factors that may be responsible for the remediation technology selection are taken into account, including site conditions, environmental factors, technical factors, and economic factors [80]. Although several remediation technologies may be available for a contaminated site, none can be applied universally, since every remediation technology has its own advantages and disadvantages. Therefore, the application of a screening method could be more time-saving and cost-saving for selecting the most suitable remediation technology for practical use. Furthermore, a practical remediation program at an OCP-contaminated site is used to illustrate the effectiveness of this selection method.

## 2. Materials and Methods

### 2.1. OCPs-Contaminated Site Investigation

#### 2.1.1. Site Characterization

A DDT manufacturer produced a large number of pesticide products in the 1980s. The floor space was 130,000 m^2^. The surrounding area of the plant was a residential area. The original enterprises in the plot have been shut down and all of the buildings have been demolished.

#### 2.1.2. Soil Sampling

Soil sampling was performed on the basis of the Technical Guidelines for Environmental Site Monitoring (HJ25.2-2014, http://kjs.mee.gov.cn/hjbhbz/bzwb/jcffbz/201402/t20140226_268360.shtml). Soil samples were collected at a depth of 0–10 cm from boreholes at different locations, including the site of the former DDT production workshop, the site of the former package workshop, sites that were both long and short distances from the DDT production areas, and also sites along the main transportation corridor. Moreover, several potential heavily contaminated sites, such as production workshops, sewage pipelines, waste-to-stacking, etc., were selected as the monitoring block according to our investigation and interview. At each site, five subsamples from different randomly placed subplot points were taken for one composite sample.

#### 2.1.3. Soil Pretreatment

The soil samples were air-dried, and sieved by a steel mesh. The soil samples were spiked with surrogate standards and then Soxhlet-extracted with *n*-hexane/acetone mixture. The extracts were concentrated by rotary vacuum evaporation, and then solvent-exchanged to *n*-hexane. Cleanup was performed using a silica gel column. Before the extract was loaded, anhydrous sodium sulfate was added at the top and then pre-eluted by *n*-hexane. After the addition of extract, the column was eluted with *n*-hexane followed by *n*-hexane/dichloromethane mixture. The two fractions were combined as a single fraction and then concentrated under a stream of nitrogen before analysis. Both matrix blanks and method blanks were analyzed with five samples.

#### 2.1.4. Instrumental Analysis

DDT were quantified by an Agilent 6890 gas chromatograph with electron capture detector (GC-ECD, Agilent Technologies Inc., Palo Alto, CA, USA). The temperature program was as follows: initial temperature 100 °C held for 2 min, increased to 200 °C at a rate of 5 °C/min and held for 2 min, then ramped up to 280 °C at a rate of 8 °C/min and maintained for 10 min.

The results of the OCP-contaminated site investigation showed that DDT was mainly distributed in the soil from the surface to 5 m below the ground, and there was no clear difference in the concentration of pollutants between the soil layers. Among the sites, the area near the original DDT production workshop and warehouse was more polluted, and the concentration of DDT in the soil was close to 1000 mg/kg. This may be the result of backward production techniques and imperfect environmental protection facilities at this location having led to the leakage of DDTs, which may have resulted in high levels of DDT residues in the process of product production and goods stacking (such as raw materials, and both semi-finished and finished products). The soil was mainly silty clay, and the surface layer was mixed soil. The total contaminated soil was about 300,000 m^3^.

#### 2.1.5. Remediation Goal

According to the future land use and redevelopment requirements of the site, it was initially determined that the remediation target was DDTs ≤ 1 mg/kg. Since the given remediation time was short, it was necessary to select a remediation technology with a rapid, high removal rate and low operating costs.

### 2.2. Screening Process

#### 2.2.1. Establishment of the Hierarchical Analysis Model

On the basis of the principle of hierarchical analysis, a model was proposed according to the investigation results of the OCP-contaminated site, including target layer (A), criterion layer (B). and indicator layer (C). In the criterion layer, there are three indicators, which are economic indicator (B1), environmental indicator (B2), and technical indicator (B3). Notably, environmental impact assessment is according to three factors, which are degree of difficulty of land reuse after remediation (C4), residual pollution (C5), and harm of by-products (C6). The hierarchical analysis model is shown in Figure 1.

#### 2.2.2. Preliminary Screening of Remediation Technology

In this paper, the preliminary screening of remediation technology was the selection of potentially usable remediation technologies from existing remediation technologies according to the characteristics of DDT. According to the contaminated site conditions and the technical applicability, the potential technologies for consideration include chemical reduction (D1), soil leaching (D2), thermal desorption (D3), and co-processing in cement kiln (D4). Supporting information for the remediation technology options is shown in Table 1.

#### 2.2.3. Consistency Evaluation

The relative weights of criteria are calculated by pairwise comparisons. The linguistic scale consists of linguistic terms and a number between one and nine associated with this linguistic variable. The relative importance values of the indices (from 1 to 9) are shown in Table 2.

The maximum eigenvalue λ_max_ of each judgment matrix and its corresponding eigenvector ω are calculated. Then all matrices are put through a consistency check. If it passes the test, the eigenvector corresponding to the largest eigenvalue is the weight. If any inconsistency is detected, then the matrix is formed again. The consistency index is calculated as follows:CI = (λ_max_ − *n*)/(*n* − 1)(1)
where *n* is the order of the judgment matrix.

A random consistency indicator RI is introduced to measure the size of CI; an average random consistency index RI is shown in Table 3. The consistency ratio CR = CI/RI, when CR < 0.1, then the matrix is considered to pass the consistency test.

#### 2.2.4. Weight Calculation

The weight calculation of the scheme layer is performed by combining the weight under the single criterion from top to bottom. Suppose that layer A contains *m* decision objectives, which are A_1_…A*_m_*, and their total ordering weights are *a*_1_, … *a_m_*, respectively. The next level B layer is further composed of n factors *B*_1_, … *B_m_*, and their hierarchical single order weights for *A_j_* are *b_1j_*, …, *b_nj_*, respectively (when B_i_ is not associated with *A_j_*, *b_ij_* = 0). Now we want to find the weight of each factor in the B layer in regard to the total goal, that is, to find the total ranking weights b_1_…b*_n_* of the factors of the B layer. Then the total ranking weight was calculated according to the following equation:(2)$$b_{i}=\sum_{j=1}^{m}b_{ij}a_{j},\;i=1,\ldots ,n$$

#### 2.2.5. Optimization by TOPSIS

Step 1: Based on the AHP of the score of the preliminary screening results, matrix A is constructed with a standardized decision as follows:(3)$$r_{ij}=\frac{a_{ij}}{\sqrt{\sum_{i=1}^{m}a_{ij}^{2}}}\;\;\;\;\;i=1,\ldots \ldots m;\;j=1,‥\ldots n$$

Step 2: Establish a weighted normative matrix *X* = {*a_ij_*}. Calculate the weight determined through AHP method ω = (ω_1_, ω_2_,…, ω*_n_*) T, and the use the following:
*v_ij_* = ω_j_·*r_ij_*, *i* = 1,…*m*; *j* = 1,…*n*(4)

Step 3: The positive and negative ideal solutions are determined. The *j*th parameter value of the ideal solution *x^*^* is defined as *x^*^_j_*, and *j*th parameter value of the *x*^0^ is defined as *x*^0^*_j_*, then the ideal solution and negative ideal solution are as follows:

Ideal solution:(5)$$d_{i}^{*}=\sqrt{\sum_{j=1}^{n}{\left(x_{ij}-x_{j}^{*}\right)}^{2}},\;i=1,‥‥m$$

Negative ideal solution:(6)$$d_{i}^{0}=\;\sqrt{\sum_{j=1}^{n}{\left(x_{ij}-x_{j}^{0}\right)}^{2}},\;i=1,‥‥m$$

Step 5: Calculate the queued indication value (the comprehensive evaluation index) of each scheme.
(7)$$c_{i}^{*}\;=\;d_{i}^{0}/\left(d_{i}^{0}+d_{i}^{*}\right),\;i=1,‥‥m$$

Step 6: Rank the $c_{i}^{*}$ in descending order. Define C_i_ between 0 and 1. The alternative with the largest $c_{i}^{*}$ value is the best choice.

## 3. Results and Discussion

### 3.1. Weight Calculation of Criterion Layer

The weight calculation results of various factors in the criterion layer are shown in Table 4. Among them, the weights of economic indicators, environmental indicators and technical indicators are 0.2290, 0.6955, and 0.0754 respectively. This shows that environmental indicators are the most important factors for consideration regarding the restoration of organochlorine-contaminated sites, followed by economic indicators, and finally technical indicators. This is similar to the screening results for remediation technologies of POPs-contaminated site reported in the literature [81].

### 3.2. Weight Calculation of Scheme Layer

The total ranking weight of the calculation scheme layer is shown in Table 5. It can be seen from Table 5 that in this case, co-processing in cement kiln has the highest weight (D4, 0.3000), followed by soil leaching (D2, 0.2915) and thermal desorption (D3, 0.2475). The weights of the two technologies are very close, and they are all ex-situ remediation techniques.

### 3.3. Calculation of the Initial Decision-Making Matrix

According to the information for the remediation technology options in Table 1, scores for each remediation technologies are determined based on TOPSIS steps. The initial decision matrix data is shown in Table 6.

The initial decision matrix data is normalized according to the algorithm of Equation (1), and a normalized matrix is obtained in Table 7.

According to the eigen-analysis method, the index weights ω = (0.2290, 0.695, 0.0754) are calculated according to Equation (4) to obtain the weighted specification matrix in Table 8.

The weighted gauge array is sequentially sorted according to Equations (3)–(7); the final sort results are shown in Table 9.

It can be concluded from the evaluation results that the scores of the four remediation technologies are between 0.1462 and 0.4647, where the priority order of the alternative technologies is D4, D3, D1, D2. Although the AHP method can also be applied for criteria scoring, it was only used to determine criteria weights in this study. The optimal remediation technology obtained from both AHP and TOPSIS was co-processing in cement kiln, indicating that the result is correct. Compared to other remediation technologies, co-processing in cement kiln is also used in the remediation projects of OCPs-contaminated sites reported in the literature [7,62,82,83]; this may also be attributed to the fact that most of the soil remediation technologies in China are still at the stage of laboratory simulation. In order to save costs and shorten the repair time for engineering practice, it is more economical and safe to choose the cement kiln co-disposal technology to obtain maximal benefits.

### 3.4. The Practical Remediation Project for an OCP-Contaminated Site

The screening results of the AHP-TOPSIS method show that the optimal technology was co-processing in cement kiln, which is similar to the practical remediation project for an OCP-contaminated site. In this project, there are mainly three soil treatment processes, including contaminated site excavation, contaminated soil transportation, and cement plant disposal. The actual disposal process is shown in Figure 2. The contaminated soil was first collected from the DDT-contaminated site. After the contaminated soil was sealed and transported to the cement plant, the contaminated soil was crushed and transported to the grinding mill. Then a pre-homogenization treatment was carried out in the raw meal homogenizing silo. Next, the contaminated soil was transferred to the pre-processor for drying at about 300–1300 ˚C, and the dried soil was transported to the preheater according to a fixed proportion of ingredients. After that, incineration with 950–1300 °C took place in the decomposition tower. The incineration procedure can prompt the DDT in the contaminated soil to degrade completely.

A summary of remediation projects of POPs-contaminated sites was provided in recent years in China. As shown in Table 10, it can be seen that co-processing in cement kiln can also be applied to other types of OCPs-contaminated sites. Since the remediation technologies for OCPs-contaminated sites are not mature in China, OCPs-contaminated sites remediation research is reported with little engineering practice. Nevertheless, co-processing in cement kiln has been widely applied for the remediation of other types of POPs-contaminated sites. Since China still lacks standards for the restoration of POPs-contaminated soils and evaluation systems, the selection of co-processing in cement kiln seems to be the best alternative for contaminated site reuse. Therefore, the current study showed that the decision-making problem of remediation technology screening for OCPs-contaminated sites can be solved by the combined AHP-TOPSIS method. Furthermore, the proposed approach can provide an efficient manner in which to help decision makers obtain reasonable and credible results for remediation technology selection in practical applications.

## 4. Conclusions

The selection of an appropriate remediation technology for an OCP-contaminated site was developed by incorporating AHP and TOPSIS. By taking a practical remediation project as an example, four suitable technologies were preliminarily selected according to their applicability to site-specific conditions. Next, a hierarchical analysis model was established that took into consideration economic factors, technical factors, and environmental factors that could affect the decision-making process. Particularly, the environmental impact of the alternative technology was the main decision parameter in the overall assessment. The AHP approach was then conducted to rank the candidate remediation technologies according to their weights. The results showed that co-processing in cement kiln had the highest weight in this case study.

TOPSIS was further applied to determine the priority order of the alternative remediation technologies. The results showed that co-processing in cement kiln was the optimum solution. The high score of the co-processing in cement kiln option was mainly due to the fact that co-processing in cement kiln can effectively remove OCPs, leading to a good score according to effect. Based on the screening results of AHP-TOPSIS, co-processing in cement kiln was suggested as the optimal remediation technology. This remediation technology was similar to the practical remediation project, indicating that the screening method could be applied for the selection of remediation technologies. Furthermore, this screening approach has great potential to help decision makers choose reasonable remediation options in practical applications for POPs-contaminated sites.

## Figures and Tables

**Figure 1 ijerph-16-01888-f001:**
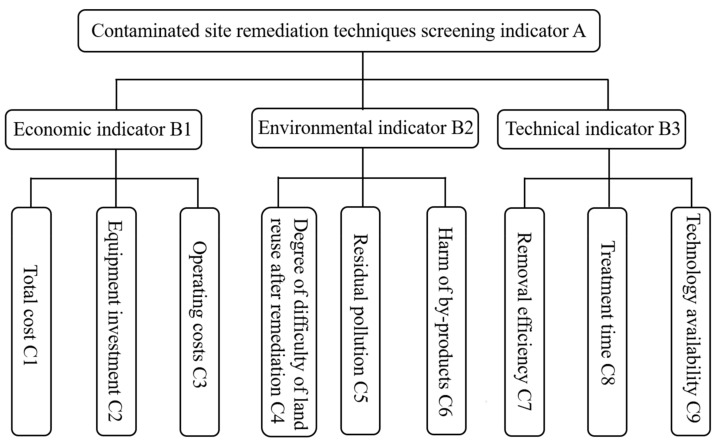
Hierarchical analysis model for selecting the remediation technologies.

**Figure 2 ijerph-16-01888-f002:**
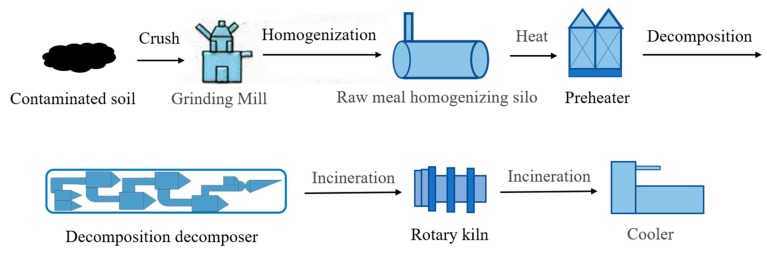
Co-processing in cement kiln for contaminated soil treatment.

**Table 1 ijerph-16-01888-t001:** Information for the remediation technology options.

Criteria	Chemical Reduction	Soil Leaching	Thermal Desorption	Co-Processing in Cement Kiln
In situ/ex-situ	In situ/Ex-situ	In situ/Ex-situ	In situ/Ex-situ	Ex-situ
Processing object	Organic Pollutants	Semi-volatile organic pollutant	Volatile pollutant	Organic Pollutants
Soil properties	Sand/high permeability	No request	Sand/Low permeability	No request
Technical maturity	More mature abroad	More mature abroad	More mature abroad	More mature internal
Cleanup time (months)	3–6	3–12	6–12	<3
Overall cost ($/t)	400–850	50–400	450	100–180
Removal rate	>90%	>80%	>99.99%	>90%
Environmental impact	Toxic by-product	Toxic by-product	Dioxin	Dioxin
Secondary pollution risk	Normal	Normal	Normal	Slight
Degree of reuse	Reusable	Reusable	Reusable	Unusable

**Table 2 ijerph-16-01888-t002:** Relative importance values of the indices.

Scaling	1	3	5	7	9
Importance of the two factors	Equal	Slightly important	Important	Very important	Extremely important
	2, 4, 6, and 8 are the intermediate values of the above adjacent judgments				

**Table 3 ijerph-16-01888-t003:** Average random consistency index RI.

*n*	1	2	3	4	5	6	7	8	9	10	11
RI	0.00	0.00	0.58	0.91	1.12	1.24	1.32	1.41	1.45	1.49	1.51

**Table 4 ijerph-16-01888-t004:** Weight coefficient of all factors in the hierarchy structure.

Criterion Layer	Indicator Layer	Weight
B1	C1	0.7928
	C2	0.1313
	C3	0.0760
B2	C4	0.1634
	C5	0.2970
	C6	0.5396
B3	C7	0.1365
	C8	0.2385
	C9	0.6250

**Table 5 ijerph-16-01888-t005:** Calculation of weight coefficients of the D layer.

Factors	D1	D2	D3	D4
C1	0.0563	0.0874	0.1983	0.6581
C2	0.0874	0.0563	0.1983	0.6581
C3	0.0914	0.0452	0.2600	0.6035
C4	0.2844	0.4729	0.1699	0.0729
C5	0.3132	0.4965	0.0509	0.1393
C6	0.1772	0.3001	0.4753	0.0475
C7	0.0439	0.0877	0.3130	0.5555
C8	0.0954	0.1601	0.2772	0.4673
C9	0.1570	0.0882	0.2720	0.4829
Total weight of D layer	0.1875	0.2915	0.2475	0.3000

**Table 6 ijerph-16-01888-t006:** Initial decision-making matrix.

Indicators	D1	D2	D3	D4
B1	2	4	3	5
B2	4	2	3	5
B3	5	3	4	3

**Table 7 ijerph-16-01888-t007:** Normalized decision-making matrix.

Indicators	D1	D2	D3	D4
B1	0.2722	0.5443	0.4082	0.6804
B2	0.5443	0.2722	0.4082	0.6804
B3	0.6804	0.2722	0.5443	0.4082

**Table 8 ijerph-16-01888-t008:** Weighted normalized decision-making matrix.

Indicators	D1	D2	D3	D4
B1	0.0623	0.1246	0.0934778	0.1558
B2	0.3783	0.1892	0.283699	0.4729
B3	0.05130	0.0205	0.04104022	0.0308

**Table 9 ijerph-16-01888-t009:** Preference order of the evaluated techniques.

Parameters	D1	D2	D3	D4
*d_i_^*^*	0.3271	1.1576	0.6052	0.544
*d_0_^*^*	0.0807	0.1982	0.3739	0.4723
*C_i_**	0.1979	0.1462	0.3819	0.4647
Rank	3	4	2	1

**Table 10 ijerph-16-01888-t010:** Remediation projects of POPs-contaminated sites in China. POPs: persistent organic pollutants; DDTs: dichlorodiphenyltrichloroethanes; HCHs: hexachlorocyclohexanes.

Province/Cities	Contaminated Sites	Contaminants	Maximum Content	Remediation Technology	Scale (10,000 m^3^)	Treatment Standard
Beijing	Paint plant	DDTs, HCHs	-	Co-processing in cement kiln	14	Standard for rural residential land
Beijing	Paint plant	HCHs, DDTs	2210 mg/kg	Co-processing in cement kiln	25.5	Site evaluation restoration target criteria
Beijing	Pesticide plant	DDTs, HCHs	-	Cure incineration	2.7	Standards for construction land for public transport hubs
Hubei	Pesticide plant	DDTs, HCHs	4661 mg/kg	Co-processing in cement kiln	29.7	Site risk assessment criteria
A southern city	Pesticide plant	HCHs, DDTs	2989 mg/kg	Co-processing in cement kiln	29.68	Site risk assessment criteria
A city (Guangzhou)	Paint plant	HCHs, DDTs	3210 mg/kg	Co-processing in cement kiln	5.3	Standard for rural residential land
A county (Hebei)	Pesticide plant	DDTs, HCHs	2098 mg/kg	Co-processing in cement kiln	0.98	Site risk assessment criteria
Tianjin	Pesticide plant	DDTs, HCHs	3012 mg/kg	Co-processing in cement kiln	3.2	Standard for rural residential land

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
