# Peer review of "Application and Selection of Remediation Technology for OCPs-Contaminated Sites by Decision-Making Methods"

_ijerph, 2019, doi:10.3390/ijerph16111888_

Round 1

Reviewer 1 Report

The manuscript titled "Application and selection of remediation technology for OCPs-contaminated sites by decision-making methods" by Tiang and co-authors is an interesting application of a combination of AHP and TOPSIS models as decision-making model for the screening of remediation techniques of organochloride pesticides contaminated sites.

The text falls within the aims of the Journal and it is well written.It is the reviewer's opinion that the manuscript should be accepted after only minor revisions, as detailed hereafter:

line 35: change is with are

line 42: replace cause with promotes

lines 43-44: the sentence is not clear, please rephrase it

line 57: remove fuzzy

line 102: m2 (apex)

line 103: add T capital letter

line 109: replace obvious with clear

line 120: remove hierarchical analysis

lines 131-133: the sentence is not clear, please rephrase it

line 135:add T capital letter

Table 2: modify the order of intermediate values (2, 4, 6 and 8)

line 149: replace W capital letter

lines 214-215: the sentence is not clear, please rephrase it

line 231: remove "the contaminated soil"

line 233: replace certain with fixed

lines 234-235: the sentence is not clear, please rephrase it

Table 10: table 10 is an adding value to the manuscript, however it has too much data inside. Please reduce it taking into account only more significative data (consider to reduce it of at least 50%).

Author Response

Point 1: line 35: change is with are.

Response 1: is” has been changed with “are”, as line 36 in the revised manuscript.

Point 2: line 42: replace cause with promotes.

Response 2: replace” has been replaced with “promotes”, as line 43 in the revised manuscript.

Point 3: lines 43-44: the sentence is not clear, please rephrase it.

Response 3: The sentence “Although the OCPs contaminated industrial site may have a smaller scale than that of an agricultural site, it can cause higher pollution” has been revised as “Although the area of OCP pollution of industrial land is smaller than that of agricultural land, the impact of pollution is greater.”, as line 44-45 in the revised manuscript..

Point 4: line 57: remove fuzzy.

Response 4: fuzzy” has been removed, as line 59 in the revised manuscript.

Point 5: line 102: m2 (apex).

Response 5: m2” has been revised as “m2, as line 110 in the revised manuscript.

Point 6: line 103: add T capital letter.

Response 6: the sampling” has been revised as “Soil sampling”, as line 113 in the revised manuscript..

Point 7: line 109: replace obvious with clear.

Response 7: obvious” has been replaced with “clear”, as line 136 in the revised manuscript.

Point 8: line 120: remove hierarchical analysis.

Response 8: hierarchical analysis” has been removed, as line 151 in the revised manuscript.

Point 9: lines 131-133: the sentence is not clear, please rephrase it.

Response 9: The sentence “The preliminary screening of remediation technology in this paper was the selection of potentially usable remediation technologies from existed remediation technologies due to characteristics of DDT.” has been revised as “The preliminary screening of remediation technology in this paper was the selection of potentially usable remediation technologies from existing remediation technologies due to characteristics of DDT.” as line 162-163 in the revised manuscript.

Point 10: line 135: add T capital letter.

Response 10: the” has been revised as “The”, as line 166 in the revised manuscript.

Point 11: Table 2: modify the order of intermediate values (2, 4, 6 and 8).

Response 11: The order of intermediate values has been modified to 2, 4, 6 and 8 as line 173 (table 2) in the revised manuscript.

Point 12: line 149: replace W capital letter.

Response 12: Where” has been revised as “where”, as line 180 in the revised manuscript.

Point 13: lines 214-215: the sentence is not clear, please rephrase it.

Response 13: The sentence “While the optimal remediation technology obtained by AHP and TOPSIS was co-processing in cement kiln, indicating that the TOPSIS is consistent with the AHP.” has been revised as “The optimal remediation technology obtained from both AHP and TOPSIS was co-processing in cement kiln, indicating that the result is correct.”, as line 234-235 in the revised manuscript.

Point 14: line 231: remove "the contaminated soil".

Response 14: Thanks for the comments. We have carefully checked the sentence and the context, and we found that if “the contaminated soil” is removed, there would be no subject in this sentence. We think “the contaminated soil” should be retained.

Point 15: line 233: replace certain with fixed.

Response 15: certain” has been replaced with “fixed”, as line 266 in the revised manuscript.

Point 16: lines 234-235: the sentence is not clear, please rephrase it.

Response 16: The sentence “this procedure can prompt DDT in the contaminated soil degraded completely.” has been revised as “The incineration procedure can prompt DDT in the contaminated soil to degrade completely.”, as line 255-256 in the revised manuscript..

Point 17: Table 10: table 10 is an adding value to the manuscript, however it has too much data inside. Please reduce it taking into account only more significative data (consider to reduce it of at least 50%)..

Response 17: In response to the reviewer’s comments, several data have been reduced. See table 10.

Reviewer 2 Report

General comments

Appropriate and well written manuscript.

Detailed comments

Line 16: Not only ‘abuse’ but also ‘use’ of OCPs in agriculture and industries can cause high levels of residues. It might, therefore, be better to start the Abstract by “The production and use of organochlorine …”.

Line 18: The phrase “At present, there have been …” is contradictory. Write “At present, there are …”

Lines 18, 23 and others: When ‘OCP’ is used as a modifier (adjective) is should be written in singular form. Hence, it is better to write “… OCP contaminated soil remediation” instead of “… OCPs contaminated soil remediation”.

Line 19: Modify the phrase “… remediation of contaminated sites may be suffered by a series of problems …” to read “… remediation of contaminated sites may suffer from a series of problems …”.

Line 24: Verb in present tense: “… screening procedure is proposed …”.

Line 26: Remove the word ‘respectively’.

Line 27: Verb in present tense: “… results show …”.

Line 54: The phrase “… increase the utilizable of contaminant soil …” is unclear. Do the authors mean “increase the usability of contaminant soil”?

Line 57: Regarding the phrase “… the fuzzy problems may be generated for remediation technology selection”: Are the problems defininte? If not, remove the definite article ( “… fussy problems may be generated …”).

Line 68: “The multiple criteria decisionmaking methods …”: Remove the definite article. There is not a specific set of multiple criteria decision-making methods.

Line 69: “… by which the alternative remediation technologies …”: There is not a specific set of alternative remediation technologies. Hence remove the defininte article (“… by which alternative remediation technologies …”.

Line 76: Write ‘decisions’ (in plural).

Line 92: Remove the two definite articles (‘the’) in “… the screening of the remediation techniques …”

Line 94: Remove ‘decisions’ (duplication).

Line 97: Double period.

Line 102: Superscript ‘2’ in unit.

Line 103: Capitalize first letter in the sentence (“The sampling …”).

Line 112: Superscript ‘3’ in unit.

Line 115: Rewrite “… target was DDTs≤1mg/kg” to read “target was ≤1 mg/kg of DDT” (including space between number and unit).

Line 123: Remove the word “respectively”. The three listed indicators do not correspond to any previously listed entities.

Line 132: “… existed …” should probably read “… existing …”?

Line 135: Remove the word “respectively”. The four listed indicators do not correspond to any previously listed entities.

Line 135: Capital first letter in sentence.

Line 136: Use verb in present tense (“… is shown in Table 1”).

Table 1: Use capital first letters consistently in the columns.

Line 148: Insert space before and after the equal sign.

Line 155: Capital first letter.

Line 155: “Weight” used as adjective should be in sigular form. Correctly written in the next sentence (The weight calculation …”) and following text.

Line 157: Is the phrase “Set a layer A layer …” correct?

Line 161: Use subscript for ‘1’ and ‘n’ as done previously.

Table 4: Maybe better to use the term “Criterion layer” (singlular modifier) in Column 1 to be consistent with the text? See Line 185 (“… criterion layer in Table 4”).

Line 199: Capital ‘t’ in ‘Table 1’.

Line 203: Capital ‘e’ in ‘Equation 1’.

Line 206: Regarding the phrase “According to the Eigen analysis method …”: The method is commonly written as a common noun, i.e., lowercase ‘e’. This is consistent with the common nouns ‘eigenvalue’ and ‘eigenvector’ used earlier in the manuscript.

Line 220: Write “… economical and safe …” (both adjectives; not one adjective and one adverb).

Line 226: “Process” should be in plural form (“… three soil treatment processes …”).

Line 232-234: Insert space between number and unit: 300 ˚C – 1300 ˚C.

Line 234: Capital first letter of sentence.

Lines 234-235: Rewrite “This procedure can prompt DDT in the contaminated soil degraded completely” to “This procedure can prompt DDT in the contaminated soil to degrade completely”.

Figure 2: The use of period after table and figure captions is not consistent. Check all tables and figures for consistency.

Lines 241-242: Use ‘research’ in singular form:  “… remediation research is reported …”.

Lines 245-246: Regarding the phrase “Therefore, it showed …”: It? The current study? If so, spell out: “Therefore, the current study showed …”.

Table 10: Check consistency of the use of capital letters.

Line 253: Write “… contaminated site remediation …” (singular modifier).

Line 257: Should ‘factor’ in “economic factor” be in plural form as in “technical factors” and “environmental factors”?

Line 258: Remove “decisions” (duplication).

Author Response

Point 1: Line 16: Not only ‘abuse’ but also ‘use’ of OCPs in agriculture and industries can cause high levels of residues. It might, therefore, be better to start the Abstract by “The production and use of organochlorine …”.

Response 1: The sentence “The abuse of organochlorine pesticides (OCPs) in agricultural and industrial application cause high levels of their residues,” has been revised as “The production and use of organochlorine pesticides (OCPs) in agricultural and industrial application cause high levels of their residues,”, as line 16 in the revised manuscript.

Point 2: Line 18: The phrase “At present, there have been …” is contradictory. Write “At present, there are …”.

Response 2: have been” has been revised as “are”, as line 18 in the revised manuscript.

Point 3: Lines 18, 23 and others: When ‘OCP’ is used as a modifier (adjective) is should be written in singular form. Hence, it is better to write “… OCP contaminated soil remediation” instead of “… OCPs contaminated soil remediation”.

Response 3: OCPs” in line 18, 23, 43, 59, 87, 93, 98, 121, 223, 226, 253 have been revised as “OCP”, as line 18, 24, 44, 62, 89, 95, 103, 152, 256, 258, 288 in the revised manuscript.

Point 4: Line 19: Modify the phrase “… remediation of contaminated sites may be suffered by a series of problems …” to read “… remediation of contaminated sites may suffer from a series of problems …”.

Response 4: The sentence “the remediation of contaminated sites may be suffered by a series of problems such as a long recovery cycle,” has been revised as “the remediation of contaminated sites may suffer from a series of problems such as a long recovery cycle,”, as line 19 in the revised manuscript.

Point 5: Line 24: Verb in present tense: “… screening procedure is proposed …”.

Response 5: was” has been revised as “is”, as line 19 in the revised manuscript.

Point 6: Line 26: Remove the word ‘respectively’.

Response 6: respectively” has been removed, as line 27 in the revised manuscript.

Point 7: Line 27: Verb in present tense: “… results show …”.

Response 7: showed” has been revised as “show”, as line 27 in the revised manuscript.

Point 8: Line 54: The phrase “… increase the utilizable of contaminant soil …” is unclear. Do the authors mean “increase the usability of contaminant soil”?

Response 8: utilizable” has been revised as “usability”, as line 56 in the revised manuscript.

Point 9: Line 57: Regarding the phrase “… the fuzzy problems may be generated for remediation technology selection”: Are the problems defininte? If not, remove the definite article ( “… fussy problems may be generated …”).

Response 9: the” has been removed, as line 58 in the revised manuscript.

Point 10: Line 68: The multiple criteria decisionmaking methods …”: Remove the definite article. There is not a specific set of multiple criteria decision-making methods.

Response 10: The” has been removed, as line 70 in the revised manuscript.

Point 11: Line 69: “… by which the alternative remediation technologies …”: There is not a specific set of alternative remediation technologies. Hence remove the defininte article (“… by which alternative remediation technologies …”.

Response 11: the” has been removed, as line 71 in the revised manuscript.

Point 12: Line 76: Write ‘decisions’ (in plural).

Response 12: decision” has been revised as “decisions”, as line 79 in the revised manuscript.

Point 13: Line 92: Remove the two definite articles (‘the’) in “… the screening of the remediation techniques …”.

Response 13: the” has been removed, as line 94 in the revised manuscript.

Point 14: Line 94: Remove ‘decisions’ (duplication).

Response 14: decisions” has been removed, as line 96 in the revised manuscript.

Point 15: Line 97: Double period.

Response 15: One period has been deleted, as line 104 in the revised manuscript.

Point 16: Line 102: Superscript ‘2’ in unit.

Response 16: m2” has been revised as “m2”, as line 110 in the revised manuscript.

Point 17: Line 103: Capitalize first letter in the sentence (“The sampling …”).

Response 17: the sampling method” has been revised as “Soil sampling”, as line 113 in the revised manuscript.

Point 18: Line 112: Superscript ‘3’ in unit.

Response 18: m3” has been revised as “m3”, as line 143 in the revised manuscript.

Point 19: Line 115: Rewrite “… target was DDTs1mg/kg to read target was 1 mg/kg of DDT (including space between number and unit).

Response 19: The sentence “… target was DDTs1mg/kghas been revised as target was 1 mg/kg of DDT, as line 146 in the revised manuscript.

Point 20: Line 123: Remove the word “respectively”. The three listed indicators do not correspond to any previously listed entities.

Response 20: respectively” has been removed, as line 154 in the revised manuscript.

Point 21: Line 132: “… existed …” should probably read “… existing …”?

Response 21: existed” has been revised as “existing”, as line 163 in the revised manuscript.

Point 22: Line 135: Remove the word “respectively”. The four listed indicators do not correspond to any previously listed entities.

Response 22: respectively” has been removed as line 166 in the revised manuscript.

Point 23: Line 135: Capital first letter in sentence.

Response 23: the” has been revised as “The”, as line 166 in the revised manuscript.

Point 24: Line 136: Use verb in present tense (“… is shown in Table 1”).

Response 24: was” has been revised as “is”, as line 167 in the revised manuscript.

Point 25: Table 1: Use capital first letters consistently in the columns.

Response 25: no request” has been revised as “No request”, “ex-situ” has been revised as “Ex-situ”, “low” has been revised as “Low”, and “unusable” has been revised as “Unusable”.

Point 26: Line 148: Insert space before and after the equal sign.

Response 26: CI=(λmax-n)/(n-1)” has been revised as “CI = (λmax - n) / (n - 1)”, as line 179 in the revised manuscript.

Point 27: Line 155: Capital first letter.

Response 27: weights” has been revised as “Weight” as line 186 in the revised manuscript.

Point 28: Line 155: “Weight” used as adjective should be in singular form. Correctly written in the next sentence (The weight calculation …”) and following text.

Response 28: Weights” in line155 have been revised as “Weight” as line 186 in the revised manuscript. “weights” in line 156 and line 168 have been revised as “weight”, as line 187 and line 200 in the revised manuscript.

Point 29: Line 157: Is the phrase “Set a layer A layer …” correct?

Response 28: The sentence “Set a layer A layer containing A1...Am”has been revised as “Suppose that layer A contains m decision objectives, that is A1...Am”, as line l88-89 in the revised manuscript.

Point 30: Line 161: Use subscript for ‘1’ and ‘n’ as done previously.

Response 30: b1...bn” has been revised as “b1...bn”, as line 193 in the revised manuscript.

Point 31: Table 4: Maybe better to use the term “Criterion layer” (singlular modifier) in Column 1 to be consistent with the text? See Line 185 (“… criterion layer in Table 4”).

Response 31: Criteria layer” has been revised as “Criterion layer” in table 4.

Point 32: Line 199: Capital ‘t’ in ‘Table 1’.

Response 32: table 1” has been revised as “Table 1”, as line 231 in the revised manuscript.

Point 33: Line 203: Capital ‘e’ in ‘Equation 1’.

Response 33: equation 1” has been revised as “Equation 1”, as line 235 in the revised manuscript.

Point 34: Line 206: Regarding the phrase “According to the Eigen analysis method …”: The method is commonly written as a common noun, i.e., lowercase ‘e’. This is consistent with the common nouns ‘eigenvalue’ and ‘eigenvector’ used earlier in the manuscript.

Response 34: Eigen” has been revised as “eigen”, as line 238 in the revised manuscript.

Point 35: Line 220: Write “… economical and safe …” (both adjectives; not one adjective and one adverb).

Response 35: safely” has been revised as “safe”, as line 253 in the revised manuscript.

Point 36: Line 226: “Process” should be in plural form (“… three soil treatment processes …”).

Response 36: process” has been revised as “processes”, as line 259 in the revised manuscript.

Point 37: Line 232-234: Insert space between number and unit: 300 ˚C – 1300 ˚C.

Response 37: 300°C-350°C” has been revised as “300 ˚C – 1300 ˚C”, as line 265 in the revised manuscript.

Point 38: Line 234: Capital first letter of sentence.

Response 38: this procedure” has been revised as “The incineration procedure”, as line 267 in the revised manuscript.

Point 39: Lines 234-235: Rewrite “This procedure can prompt DDT in the contaminated soil degraded completely” to “This procedure can prompt DDT in the contaminated soil to degrade completely”.

Response 39: The sentence “This procedure can prompt DDT in the contaminated soil degraded completely” has been revised as “This procedure can prompt DDT in the contaminated soil to degrade completely”, as line 267-268 in the revised manuscript.

Point 40: Figure 2: The use of period after table and figure captions is not consistent. Check all tables and figures for consistency.

Response 40: Periods have been added in Table 1, Table 6, Table 7, Table 8, Table 9 and Figure 2.

Point 41: Lines 241-242: Use ‘research’ in singular form: “… remediation research is reported …”.

Response 41: researches are” has been revised as “research is”, as line 275-276 in the revised manuscript.

Point 42: Lines 245-246: Regarding the phrase “Therefore, it showed …”: It? The current study? If so, spell out: “Therefore, the current study showed …”.

Response 42: The sentence “Therefore, it showed …” has been revised as “Therefore, the current study showed …”, as line 279-280 in the revised manuscript.

Point 43: Table 10: Check consistency of the use of capital letters.

Response 43: co-processing in cement kiln” has been revised as “Co-processing in cement kiln”.

Point 44: Line 253: Write “… contaminated site remediation …” (singular modifier).

Response 44: sites” has been revised as “site” as line 288 in the revised manuscript.

Point 45: Line 257: Should ‘factor’ in “economic factor” be in plural form as in “technical factors” and “environmental factors”?

Response 45: economic factor” has been revised as “economic factors”, as line 292 in the revised manuscript.

Point 46: Line 258: Remove “decisions” (duplication).

Response 46: decisions” has been removed, as line 293 in the revised manuscript.

Reviewer 3 Report

The OCPs contaminated sites remediation technology incorporating AHP as well as TOPSIS were studied in this paper.

Comments are as follows:

1.  The four remediation technologies in Table 1 should be tested in contaminated soils of DDT plants (the remediation target was DDTs≤1mg/kg) to select the best remediation technology.

2.  “factors that could affect the decisions decision making process were taken into account”(line 94). The meaning of this sentence is not clear enough.

3.   “identification, After” (line 107) Should be changed to “identification. After”

4.   “the area near the original DDT production workshop and warehouse is more polluted” (line 110).why?

5.  Sampling process, sample pretreatment, instrumental analysis should be supplemented in 2.1.1.

6.   “8, 6, 4, and 2 are the intermediate values of the above adjacent judgments” (Table 2). The meaning is vague.

Author Response

Point 1: The four remediation technologies in Table 1 should be tested in contaminated soils of DDT plants (the remediation target was DDTs1mg/kg) to select the best remediation technology.

Response 1: The aim of this paper is to obtain the most appropriate remediation technology on the basis of the decision-making method. The four remediation technologies can be applied for OCP contamination site remediation. Since every remediation technology has its own advantages and disadvantages, none could be universally applied to all contaminated types. Therefore, we think there may be more time-saving and economical to select one remediation technology by scientific and reasonable screening method instead of testing four remediation technologies in contaminated soils of DDT plants. Moreover, the date of the practical program provided in this study further confirmed these results.

The sentence “Although several remediation technologies may be available for a contamination site, none can be applied universally since every remediation technology has its own advantages and disadvantages. Therefore, the application of screening method could be more time-saving and cost-saving to select the most suitable remediation technology for practical use.” has been added in the introduction section, as line 99-103 in the revised manuscript.

Point 2: factors that could affect the decisions decision making process were taken into account” (line 94). The meaning of this sentence is not clear enough.

Response 2: The sentence “factors that could affect the decisions decision making process were taken into account” has been revised as “several factors, which may be responsible for the remediation technology selection are taken into account.”, as line 96 in the revised manuscript.

Point 3: identification, After” (line 107) Should be changed to identification. After”.

Response 3: After” has been removed, as line 135 in the revised manuscript.

Point 4: the area near the original DDT production workshop and warehouse is more polluted (line 110). why?

Response 4: The high contamination of the DDT in the original DDT production workshop and warehouse was mainly resulted from the long-tern backward production technology and imperfect environmental protection facilities, which may cause DDTs leakage during the production process and goods stacking such as raw material, semi-finished and finished products. Therefore, the sentence “This may contribute to the backward production technique and imperfect of environmental protection facilities, leading to leakage of DDTs, which may result in high level of DDT residues in the process of product production and goods stacking such as raw material, semi-finished and finished products.” has been added, as line 139-142 in the revised manuscript.

Point 5: Sampling process, sample pretreatment, instrumental analysis should be supplemented in 2.1.1.

Response 5: In response to the reviewer’s comments, “Sampling process”, “Sample pretreatment”, “instrumental analysis” has been provided in section 2.1 of the revised manuscript.

Point 6: 8, 6, 4, and 2 are the intermediate values of the above adjacent judgments” (Table 2). The meaning is vague.

Response 6: The order of intermediate values has been modified to 2, 4, 6 and 8.

Round 2

Reviewer 3 Report

This paper can be published.